

# Pregnancy-associated plasma protein A in maternal serum for predicting early gestational diabetes mellitus: a systematic review and meta-analysis

Yuge Gao[1], Jihong Chen[1] and Jia Mi[2]

[1] Changchun University of Chinese Medicine, Changchun, Jilin, China
[2] Affiliated Hospital, Changchun University of Chinese Medicine, Changchun, Jilin, China

Corresponding author
Jia Mi, mijia0101@126.com

## ABSTRACT

**Objective.** This study assessed the effectiveness of pregnancy-associated plasma protein A (PAPP-A) in maternal serum in predicting gestational diabetes mellitus (GDM) in different geographical regions.

**Methods.** PubMed, Web of Science, EMBASE, and Cochrane Library databases were searched for articles assessing the diagnostic utility of PAPP-A for GDM up to December 3, 2024. Two independent researchers screened the literature. Study quality was appraised using the QUADAS-2 instrument and RevMan 5.4 software. Stata 15.1 software was leveraged to calculate the combined sensitivity, specificity, positive likelihood ratio, negative likelihood ratio, diagnostic odds ratio, and heterogeneity. The summary receiver operating characteristic curve was plotted to calculate the area under the curve (AUC). Subgroup analysis was also conducted to determine the source of heterogeneity.

**Results.** This study included eight cohort studies, one cross-sectional study, and three case-control studies. A total of 25,183 participants were involved, including 5,993 GDM patients and 19,190 non-GDM patients. Deek's test suggested no notable publication bias ($P = 0.400$). All included studies were rated as "low risk" or "unclear". The combined sensitivity was 0.71 (95% CI 0.60–0.80), specificity was 0.62 (95% CI [0.55–0.68]), positive likelihood ratio was 1.9 (95% CI [1.4–2.5]), negative likelihood ratio was 0.47 (95% CI [0.30–0.72]), and diagnostic odds ratio was 4 (95% CI [2–8]). Subgroup analysis found that sample size might be the source of heterogeneity ($p < 0.05$). The AUC was 0.70 (95% CI [0.65–0.73]).

**Conclusions.** PAPP-A has a certain diagnostic value for GDM and is helpful for more accurately identifying GDM and reducing the risk of related chronic diseases. Future articles with larger sample sizes and across multiple centers are warranted to explore the combined application of PAPP-A with other biomarkers. This approach may enhance the accuracy of GDM diagnosis. Registry number: PROSPERO (CRD42024617462).

## INTRODUCTION

Gestational diabetes mellitus (GDM) is characterized by elevated blood glucose levels first detected during pregnancy, which are above the normal ranges but do not reach overt diabetes. Globally, around 14% of pregnant women suffer from GDM (*Sweeting et al., 2024*), which threatens the health of pregnant women and may lead to long-term adverse outcomes in fetuses and newborns (*He, Yang & Wu, 2024*). These consequences encompass large birth weight, neonatal hypoglycemia, breathing difficulties, and a higher likelihood of developing metabolic syndrome and type 2 diabetes (T2DM) later in life (*Senthil Kumar, Mehboob & Lei, 2024*). Approximately 60% of GDM women may develop T2DM within 5- to 10-years after childbirth (*Borna et al., 2023*). The standard approach to GDM diagnosis is an oral glucose tolerance test (OGTT) (*American Diabetes Association Professional Practice Committee, 2024*). According to the International Association of Diabetes and Pregnancy Study Groups (IADPSG), 75-gram anhydrous OGTT is recommended during 24 to 28 weeks of pregnancy. The established diagnostic criteria include fasting plasma glucose $\geq$ 5.1 mmol/L, 1-hour post-prandial plasma glucose $\geq$ 10.0 mmol/L, and 2-hour post-prandial plasma glucose $\geq$ 8.5 mmol/L (*Metzger et al., 2010*). Since this standard covers the diagnosis of various diseases and the evaluation of multiple physiological parameters, GDM diagnosis has become complex, which hinders rapid identification and diagnosis in clinical settings. Consequently, identifying effective indicators for rapid GDM diagnosis remains a significant challenge (*Cui et al., 2023*).

Pregnancy-associated plasma protein A (PAPP-A) is a glycoprotein synthesized by syncytiotrophoblast cells (*Chen et al., 2025*), with an approximate molecular mass of 200 kDa. This glycoprotein is primarily expressed in placental tissues and certain specialized cell types. It belongs to the metzincin superfamily of metalloproteinases and exhibits zinc-binding characteristics (*Lovati et al., 2013*). It mainly regulates the activity of insulin-like growth factor (IGF) and enhances endometrial receptivity, which is essential for the healthy development of the fetus (*Visconti et al., 2019*). Throughout pregnancy, PAPP-A levels rise gradually as gestational age advances (*Ramezani et al., 2020*). Recent research has increasingly recognized the potential diagnostic value of PAPP-A in GDM. However, there are notable discrepancies in its diagnostic performance. For instance, one study found that PAPP-A demonstrated strong diagnostic effectiveness, with an area under the curve (AUC) of 0.885 (*Yildiz et al., 2023*), whereas another study reported significantly lower accuracy, with an AUC of 0.479 (*Visconti et al., 2019*). Given these inconsistent findings, further research is required to determine the applicability and diagnostic precision of PAPP-A across different populations (*Ren et al., 2020*; *Sovio et al., 2024*).

Diagnostic meta-analysis is a reliable method for integrating data from various studies to precisely evaluate the sensitivity and specificity of diagnostic tests (*Bossuyt et al., 2003*). This approach effectively resolves discrepancies among research findings. In this study, we aim to integrate data to evaluate the utility of PAPP-A in the early diagnosis of GDM, thereby enhancing early detection and intervention strategies for GDM. This will provide evidence-supported guidance for clinical practice and offer evidence-based insights for

healthcare professionals and public health policymakers while identifying potential avenues for future research.

## MATERIALS AND METHODS

### Literature search

This study was registered in PROSPERO (CRD42024617462) (*Page et al., 2021*). The research followed the Preferred Reporting Items for Systematic Reviews and Meta-Analyses of Diagnostic Test Accuracy (PRISMA-DTA) statement. A detailed checklist for reporting criteria can be found in Appendix S1.

Two researchers systematically searched PubMed, Web of Science, EMBASE, and Cochrane databases for articles up to December 3, 2024. The search strategy combined subject headings and free words, and the specific strategies are detailed in Table S1. Furthermore, additional manual searches were performed to identify relevant references and similar studies related to the included papers.

#### Inclusion and exclusion criteria

Eligible studies were included according to the Population, Exposure, Comparator, Outcome, and Study (PECOS) criteria: (1) Participants: general population of suspected GDM with no limitation on gestational weeks or age (*ACOG Practice Bulletin No. 190: Gestational Diabetes Mellitus, 2018*); (2) Exposure: measurement data of PAPP-A; (3) Outcomes: true positive (TP), false positive (FP), true negative (TN), and false negative (FN); (4) A clear diagnostic gold standard; (5) Study design: cohort studies, cross-sectional studies, and case-control studies. The search was limited to English peer-reviewed journal articles.

Exclusion criteria were as follows: (1) non-English studies; (2) animal studies; (3) studies that did not calculate the PAPP-A cutoff value related to the risk factors of pregnancy syndrome; (4) studies based on the same survey/research data to avoid data duplication; (5) other literature types such as study protocols, editorials, and abstracts or conference communications. Two authors (Yuge Gao and Jia Mi) independently evaluated the titles, abstracts, and full texts. The third author (Jihong Chen) resolved the differences between the two evaluators and made the final decision.

#### Data extraction

Two researchers independently extracted data from each eligible study using a pre-designed standardized electronic spreadsheet. They documented the first author, publication year, research location (country or region), research design, gold standard, sample size, gestational weeks, subject age, threshold criteria, detection methods, sensitivity, and specificity metrics. For studies that did not directly report sensitivity and specificity, these two researchers independently calculated these values using the formulas provided in Table S1. Any differences were addressed through discussion until a consensus was achieved.

*Quality assessment*

Study quality was independently appraised by Jia Mi and Yuge Gao utilizing Review Manager 5.4 based on the QUADAS-2 tool (*Whiting et al., 2011*) in several domains, including patient selection, the test under evaluation, the reference standard, and the process flow and timing. In the QUADAS-2 assessment, if all questions were answered positively, both the risk of bias and applicability domains were rated as "Low". By contrast, if one or more questions were answered negatively, these domains were labeled as "High". In addition, if any questions were labeled as "Unclear", the risk of bias and applicability domains were classified as "Unclear". Each field was assessed for potential bias, and specific questions were asked to accurately determine this risk. Any discrepancies were addressed through discussion until an agreement was reached.

*Statistical analysis*

The sensitivity, specificity, positive likelihood ratio (PLR), negative likelihood ratio (NLR), diagnostic odds ratio (DOR), AUC, and 95% confidence interval (CI) were calculated to systematically evaluate the accuracy of PAPP-A in GDM diagnosis. Heterogeneity across studies was judged using the Cochrane $I^2$ statistic. Significant heterogeneity was indicated when either $P < 0.100$ or $I^2 > 50\%$. When significant heterogeneity was detected, a random-effects model was selected, and subgroup analyses were conducted to explore potential heterogeneity sources. Funnel plots were used for qualitative evaluation, while Deeks' test was applied for quantitative analysis of publication bias, with $P < 0.05$ indicating notable publication bias. All statistical analyses were performed in STATA 15.1. The DOR represents the ratio of the odds of developing the disease in individuals who test positive to those who test negative. This metric ranges from 0 to positive infinity, with a greater value indicating more robust discriminatory power (*Glas et al., 2003*). During meta-analysis, an appropriate effect model was selected according to heterogeneity. This approach ensured precise calculation of the combined effect size and the AUC. Furthermore, a sensitivity analysis was performed to prove the reliability and stability of the results obtained.

## RESULTS

After database searches, we initially retrieved 3,587 relevant articles. After eliminating duplicates and thoroughly assessing the titles and abstracts, we selected 39 articles for full-text evaluation. During this process, 19 articles were ruled out because they failed to meet our inclusion criteria. Finally, 20 studies were enrolled in the systematic review. Due to inadequate diagnostic outcome data, eight studies were later removed. As a result, 12 studies were incorporated into the meta-analysis (*Lovati et al., 2013*; *Syngelaki et al., 2015*; *Talasaz et al., 2018*; *Visconti et al., 2019*; *Ramezani et al., 2020*; *Ren et al., 2020*; *Kapustin et al., 2022*; *Borna et al., 2023*; *Cui et al., 2023*; *American Diabetes Association Professional Practice Committee, 2024*; *Amini et al., 2024*; *Lu et al., 2024*). The research screening process is shown in the PRISMA flowchart (Fig. 1).

This research encompassed four retrospective cohort studies, four prospective cohort studies, one cross-sectional study, and three case-control studies. These studies involved 25,183 adult participants from various countries in Asia and Europe, including

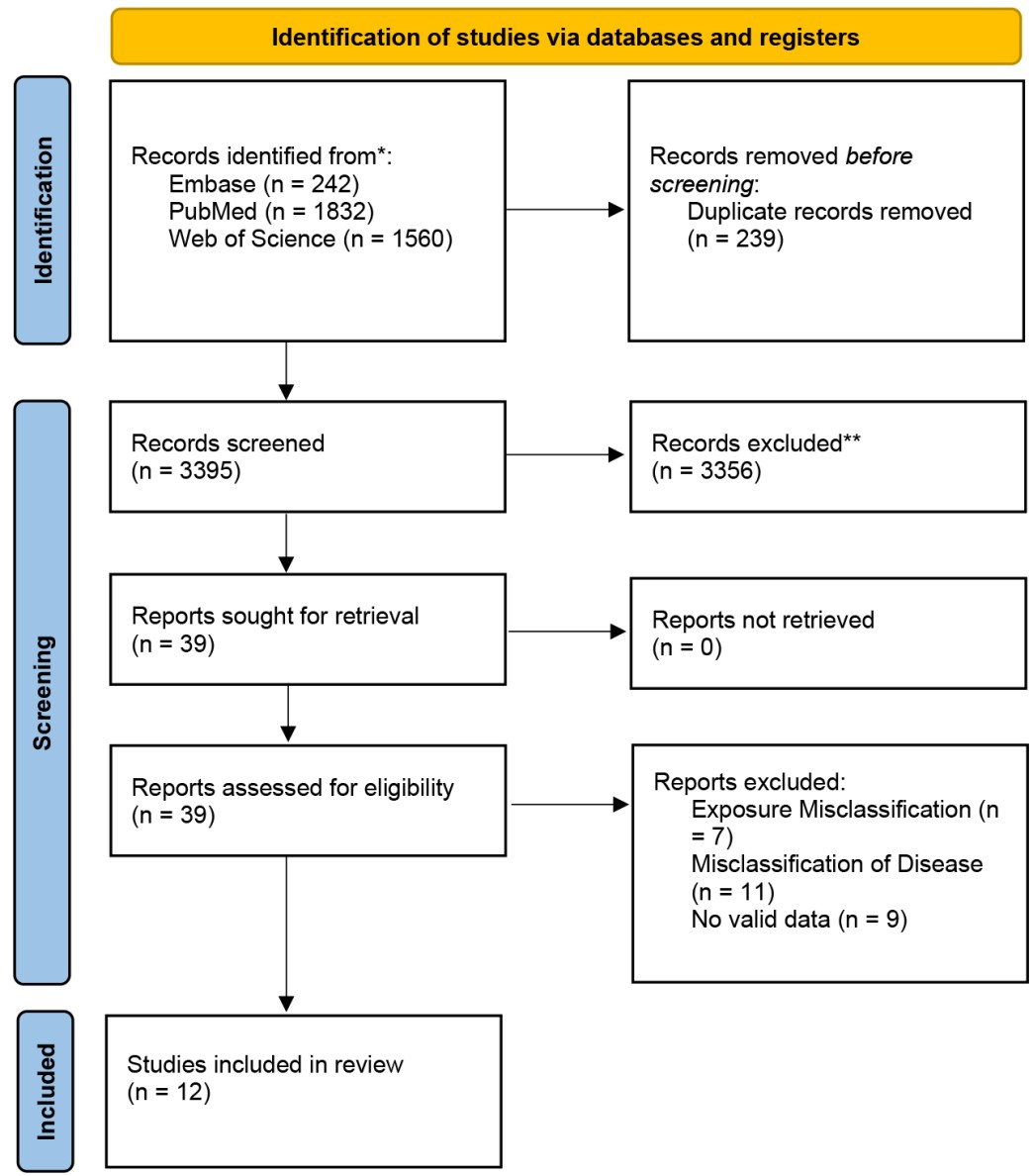

**Figure 1** **PRISMA flow chart for literature search and research selection.** Note: *Consider, if feasible to do so, reporting the number of records identified from each database or register searched (rather than the total number across all databases/registers). **If automation tools were used, indicate how many records were excluded by a human and how many were excluded by automation tools.

China, Iran, Israel, Italy, Bulgaria, and Turkey, with sample sizes varying between 99 and 7,936 individuals. The detailed study features and diagnostic outcomes are presented in Table 1. Among these studies, three provided specific PAPP-A threshold values: 4.95 IU/L (international units per liter), 16.34 ng/L (nanograms per liter), and 1.39 mg/L (milligrams per liter). The other nine studies reported PAPP-A cutoffs using median multiples of the median (MoM), ranging from 0.32 MoM to 2.21 MoM, as outlined in Table 1.

**Table 1  Main characters of the studies included in the meta-analysis.**

| Authors | Year | Area | Study design | Participants | Reference standard | Cut-off | Testing method | TP | FP | FN | TN |
|---|---|---|---|---|---|---|---|---|---|---|---|
| Yu-Ting Lu | 2024 | China | Retrospective cohort study | 1,452 | OGTT | 4.95 IU/L | TRFIA | 54 | 664 | 42 | 692 |
| Sedigheh Borna | 2023 | Iranian | Prospective cohort study | 5,854 | OGTT | 0.4MoM | Referral laboratory | 474 | 2,388 | 415 | 2,577 |
| Kinneret Tenenbaum-Gavish | 2020 | Israel | Prospective observational study | 205 | GCT/OGTT | 0.93MoM | TRFIA | 13 | 43 | 7 | 142 |
| Federica Visconti | 2019 | Italy | Retrospective cohort study | 2,424 | OGTT | 1MoM | Hexokinase method | 311 | 936 | 285 | 892 |
| Elisabetta Lovati | 2013 | Italy | Case–control study | 673 | OGTT | 0.62MoM | TRFIA | 193 | 166 | 114 | 200 |
| Vesselina Evtimova Yanachkova | 2022 | Bulgaria | Retrospective, case-control, observational study | 662 | OGTT | 0.93 MoM | ECLIA | 236 | 116 | 176 | 134 |
| Somayeh Ramezani | 2020 | Iran | Prospective analytic study | 284 | OGTT | 0.32MoM | ELISA | 147 | 35 | 54 | 48 |
| Jinhui Cui | 2023 | China | Case-control, observational study | 4,872 | OGTT | 0.83 MoM | TRFIA | 432 | 2,016 | 318 | 2,106 |
| Aşkin Yildiz | 2023 | Turkey | Retrospective cohort study | 378 | OGTT | 0.885. | FRFIA | 138 | 59 | 69 | 112 |
| Zhuo Ren | 2020 | China | Case control study | 99 | OGTT | 16.340 mIU/L | TRFIA | 37 | 8 | 14 | 40 |
| Maedeh Amini | 2024 | Iran | Cross-sectional study | 344 | OGTT | 7 MoM | TRFIA | 113 | 42 | 10 | 179 |
| Xia Wang | 2024 | China | Retrospective cohort study | 7,936 | OGTT | 1.39 mg/L | ELISA | 1,984 | 1,559 | 156 | 4,237 |

**Notes.**

TP, True Positive; FP, False Positive; FN, False Negative; TN, True Negative; OGTT, oral glucose tolerance test; GCT, Glucose Challenge Test.

*Lu et al., 2024; Borna et al., 2023; Tenenbaum-Gavish et al., 2020; Visconti et al., 2019; Lovati et al., 2013; Yanachkova et al., 2022; Ramezani et al., 2020; Cui et al., 2023; Yildiz et al., 2023; Ren et al., 2020; Amini et al., 2024; Wang et al., 2024.*

## Quality assessment

All studies were rated as either "low risk" or "unclear". Regarding patient selection, five studies were classified as "unclear", whereas seven were deemed to have "low risk". For the process and timing, all articles had "low risk of bias". For the indicator test, most studies were rated as "low bias risk". Despite potential biases in the design and execution of these studies, the overall bias risk remained relatively low (Fig. 2).

## Pooled effect size

This meta-analysis evaluated the diagnostic accuracy of PAPP-A for gestational diabetes mellitus (GDM), revealing that its overall diagnostic capability was limited, with an area under the curve (AUC) of 0.70 (95% CI [0.65–0.73]). The sensitivity was 0.71 (95% CI [0.60–0.80]), and the specificity was 0.62 (95% CI [0.55–0.68]). Subgroup analysis indicated that studies with smaller sample sizes (<1,000) demonstrated better diagnostic performance (AUC = 0.76, 95% CI [0.72–0.80]) than those with larger sample sizes. Additionally, the time-resolved fluorescence immunoassay (TRFIA) method exhibited superior performance (AUC = 0.77, 95% CI [0.73–0.80]) to non-TRFIA methods. The pooled sensitivity was 0.71 (95% CI [0.60–0.80]), specificity was 0.62 (95% CI [0.55–0.68]), AUC was 0.70 (95% CI [0.65–0.73]), positive likelihood ratio (PLR) was 1.9 (95% CI [1.4–2.5]), negative likelihood ratio (NLR) was 0.47 (95% CI [0.30–0.72]), and diagnostic odds ratio (DOR) was 4 (95% CI

a

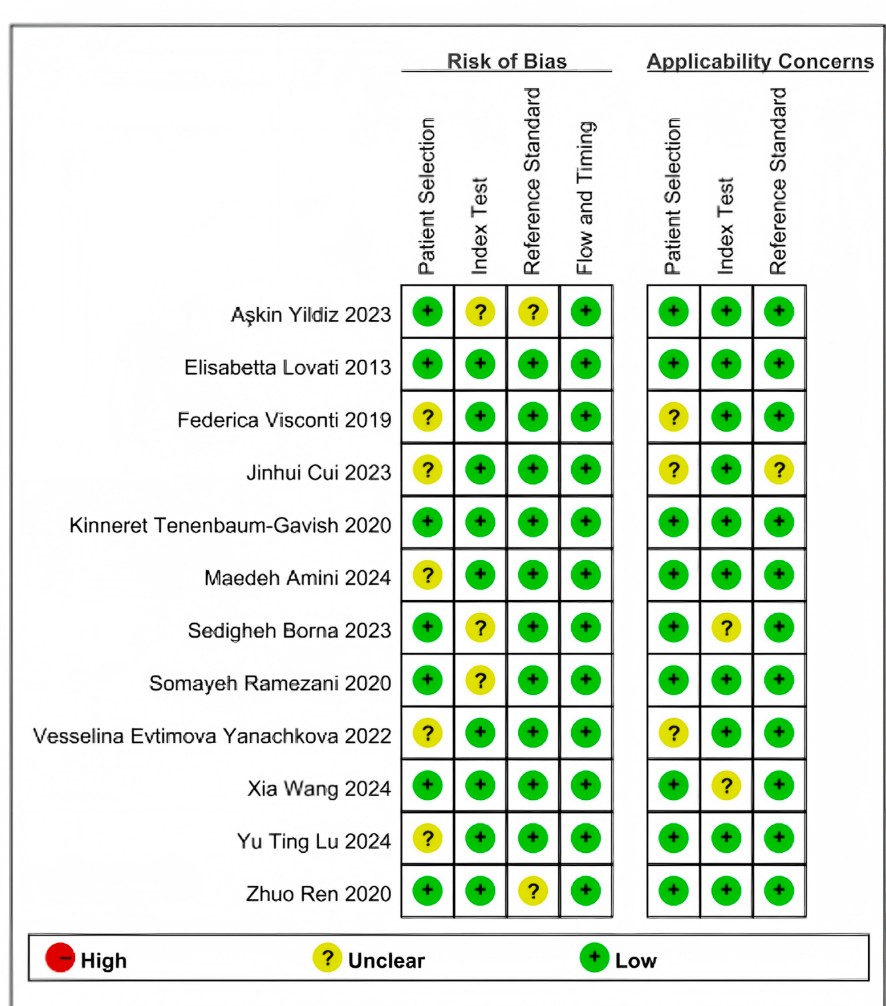

b

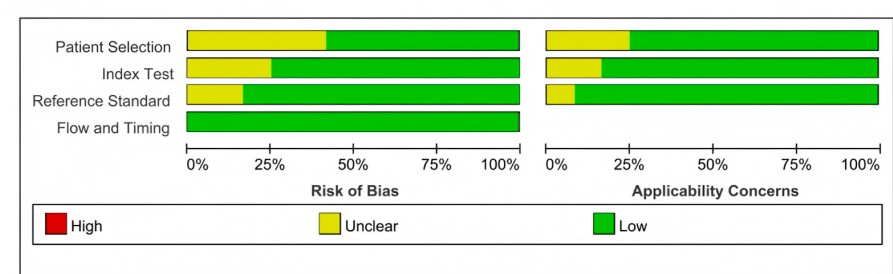

**Figure 2 Results of literature quality assessment.** (A) Domains and overall risk of bias; (B) Weighted bar chart of risk of bias assessment. *Lu et al., 2024*; *Borna et al., 2023*; *Tenenbaum-Gavish et al., 2020*; *Visconti et al., 2019*; *Lovati et al., 2013*; *Yanachkova et al., 2022*; *Ramezani et al., 2020*; *Cui et al., 2023*; *Yildiz et al., 2023*; *Ren et al., 2020*; *Amini et al., 2024*; *Wang et al., 2024*.

**Table 2** PAPP-A diagnostic analysis for GDM: overall and subgroup assessments.

| Population | Studies (n) | Sensitivity | Specificity | AUSROC | LR+ | LR- | dOR | Publication bias (p-value) |
|---|---|---|---|---|---|---|---|---|
| Overall | 12 | 0.71 [0.60, 0.80] | 0.62 [0.55, 0.68] | 0.70 [0.65, 0.73] | 1.9 [1.4, 2.5] | 0.47 [0.30, 0.72] | 4 [2, 8] | 0.400 |
| subgroup | | | | | | | | |
| sample size >1,000 | 5 | 0.66 [0.46, 0.82] | 0.56 [0.47, 0.64] | 0.61 [0.57, 0.65] | 1.5 [0.9, 2.4] | 0.61 [0.31, 1.23] | 2 [1, 8] | |
| sample size <1,000 | 7 | 0.74 [0.64, 0.81] | 0.68 [0.59, 0.75] | 0.76 [0.72, 0.80] | 2.3 [1.6, 3.3] | 0.39 [0.25, 0.60] | 6 [3, 13] | |
| TRFIA | 6 | 0.69 [0.53, 0.82] | 0.58 [0.51, 0.64] | 0.64 [0.60, 0.68] | 1.6 [1.1, 2.4] | 0.53 [0.30, 0.96] | 3 [1, 8] | |
| Non TRFIA | 6 | 0.71 [0.60, 0.80] | 0.62 [0.55, 0.68] | 0.70 [0.65, 0.73] | 1.9 [1.4, 2.5] | 0.47 [0.30, 0.72] | 4 [2, 8] | |
| Published ≤2020 | 5 | 0.67 [0.57, 0.75] | 0.65 [0.53, 0.75] | 0.71–0.67, 0.75] | 1.9 [1.2, 2.9] | 0.51 [0.34, 0.77] | 4 [2, 9] | |
| Published >2020 | 7 | 0.73 [0.55, 0.86] | 0.60 [0.52, 0.68] | 0.68 [0.63, 0.72] | 1.8 [1.2, 2.8] | 0.45 [0.22, 0.92] | 4 [1,13] | |
| case–control study | 4 | 0.63 [0.54, 0.70] | 0.60 [0.46, 0.72] | 0.65 [0.61, 0.69] | 1.6 [1.0, 2.4] | 0.63 [0.43, 0.92] | 2 [1, 6] | |
| Non case–control study | 8 | 0.75 [0.60, 0.85] | 0.63 [0.55, 0.71] | 0.72 [0.68, 0.76] | 2.0 [1.4, 3.0] | 0.40 [0.22, 0.74] | 5 [2, 14] | |

**Notes.**
AUSROC, Area under summary receiver operating characteristic curve; CI, confidence interval; dOR, diagnostic odds ratio; LR+, positive likelihood ratio; LR−, negative likelihood ratio; PAPP-A, pregnancy-associated plasma protein-A; TRFIA, time-resolved fluorescence immunoassay.

[2–8]). Significant heterogeneity was observed in the results, with $I^2$ values exceeding 90%. These findings highlight the need for further research to establish standardized protocols and enhance the diagnostic utility of PAPP-A for GDM (Table 2, Figs. 3 and 4).

## Subgroup analysis

Subgroup analyses were implemented based on publication year, sample sizes, gestational weeks, study design, cut-off, and testing method. The diagnostic threshold was evaluated for each subgroup using PAPP-A levels. Our results suggested that the variability in sample sizes might significantly contribute to the heterogeneity observed in specificity ($p < 0.01$) (Fig. 5).

## Publication bias assessment

Deek's funnel plot illustrated that the points representing individual studies in the population exhibited an approximately symmetrical distribution. Deek's test suggested no publication bias ($p = 0.400$) (Fig. 6).

## DISCUSSION

In recent years, increasing research has examined the link between PAPP-A and GDM, but the results across studies remain inconsistent. For instance, one study reported a relatively high diagnostic efficacy for GDM with an AUC of 0.82 (*Amini et al., 2024*), whereas another study concluded a lower efficacy with an AUC of 0.542 (*Borna et al., 2023*). The present research conducted a comprehensive review and meta-analysis to thoroughly examine data from various studies, thereby offering a more reliable and unbiased assessment. The results indicated that PAPP-A exhibited diagnostic value in GDM risk among pregnant women across different geographical regions, albeit with certain limitations. Specifically, the pooled sensitivity was 0.71 (95% CI [0.60–0.80]), specificity was 0.62 (95% CI [0.55–0.68]), DOR was 4 (95% CI [2–8]), and AUC was 0.70. Although PAPP-A is promising as a diagnostic indicator for GDM, its precision needs further improvement.

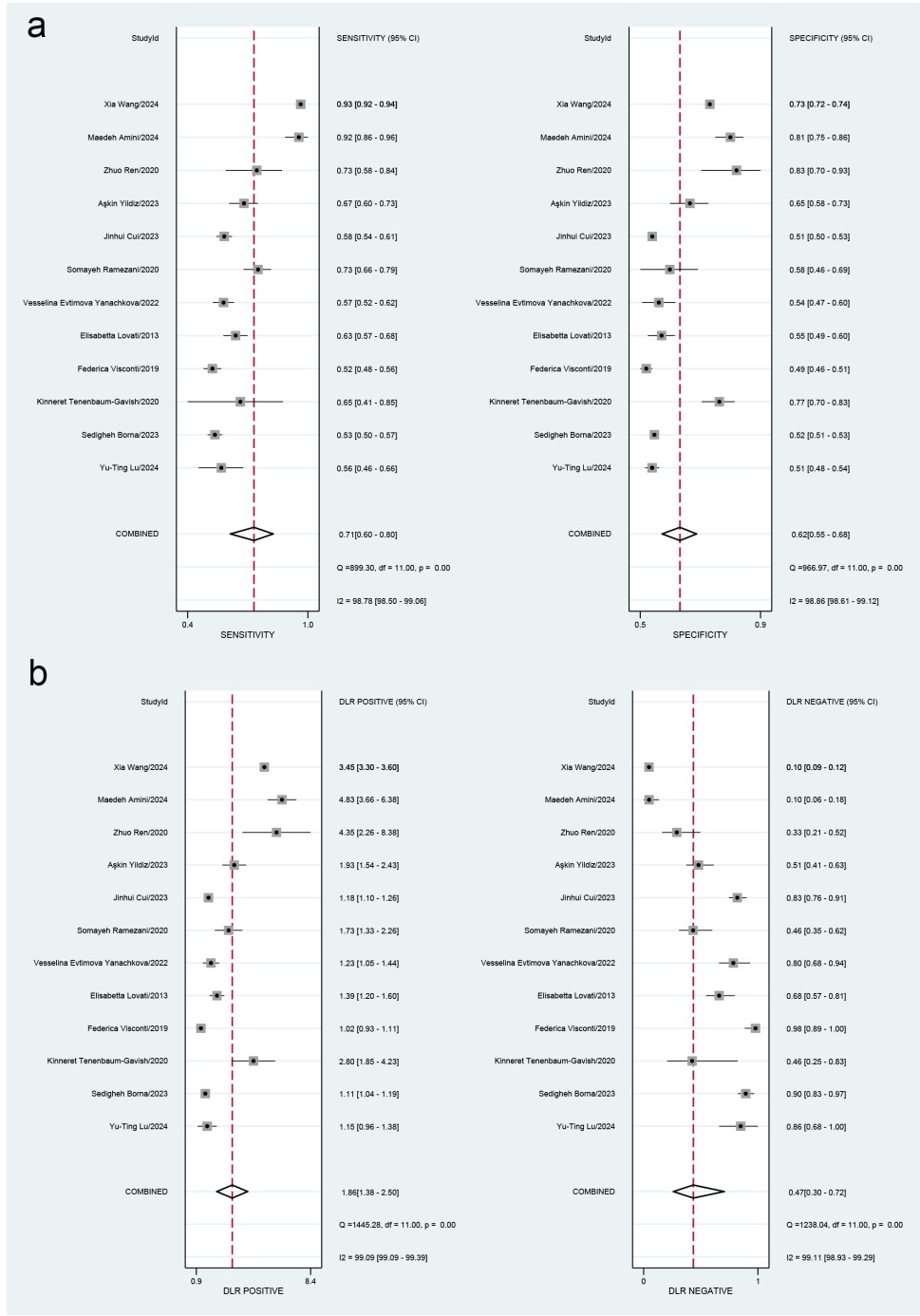

**Figure 3** (A) Forest plots for sensitivity and specificity. (B) Forest plots for the positive likelihood ratio and negative likelihood ratio. *Lu et al., 2024*; *Borna et al., 2023*; *Tenenbaum-Gavish et al., 2020*; *Visconti et al., 2019*; *Lovati et al., 2013*; *Yanachkova et al., 2022*; *Ramezani et al., 2020*; *Cui et al., 2023*; *Yildiz et al., 2023*; *Ren et al., 2020*; *Amini et al., 2024*; *Wang et al., 2024*.

The PLR of PAPP-A was 1.9 (95% CI [1.4–2.5]), indicating that a positive PAPP-A result is 1.9 times more likely in pregnant women with GDM than those without GDM,

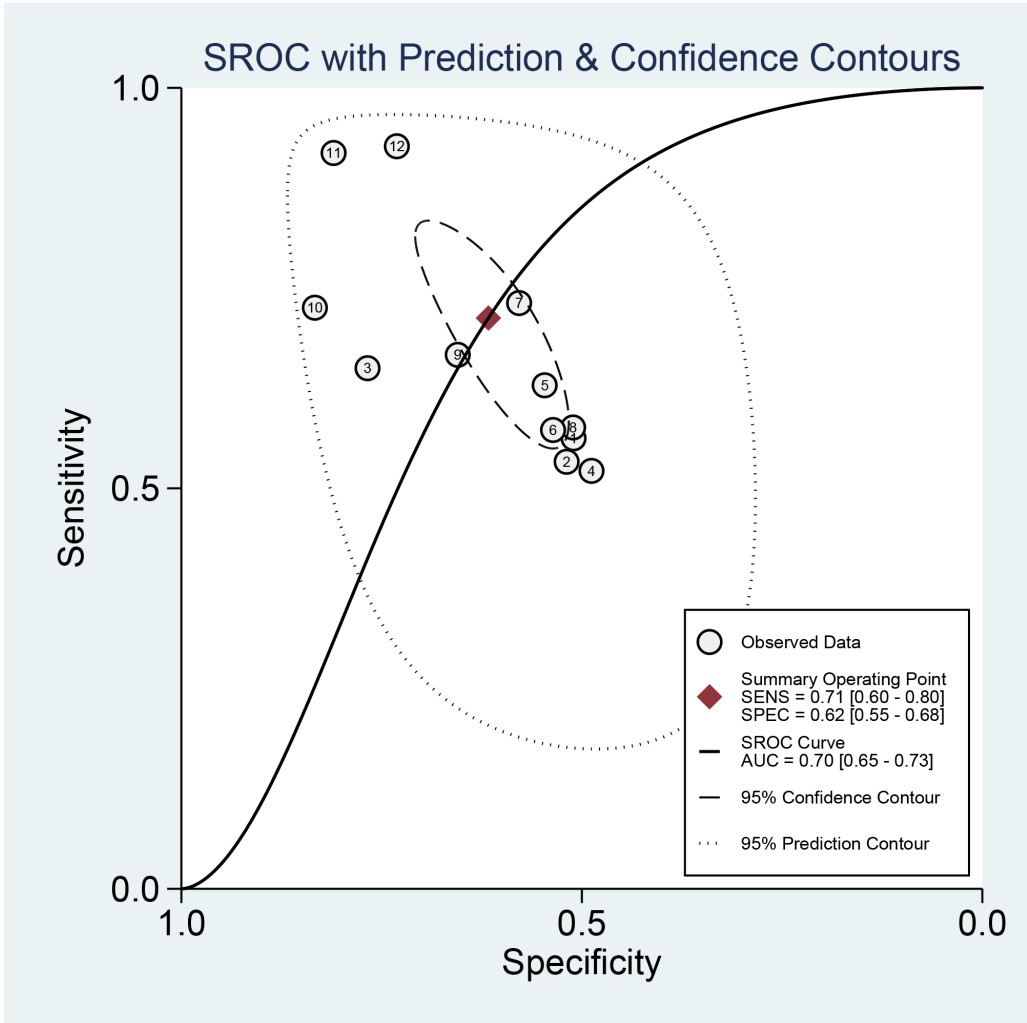

**Figure 4   Receiver operating characteristic (ROC) curve of the area under the curve (AUC) of PAPP-A for GDM diagnosis.**

highlighting its role in identifying high-risk pregnant women and prompting further diagnostic testing. The NLR was 0.47 (95% CI [0.30–0.72]), less than 1, suggesting a negative PAPP-A result can effectively rule out GDM, helping to minimize unnecessary diagnostic procedures for low-risk individuals. While PAPP-A alone has moderate diagnostic accuracy (AUC = 0.67), its PLR and NLR offer meaningful support for GDM screening.

In recent years, the worldwide increasing incidence of obesity and changes in lifestyle patterns have resulted in a significant rise in GDM incidence (*Eades, Cameron & Evans, 2017*), which differs considerably across the globe, largely influenced by the specific diagnostic standards applied. The Carpenter and Coustan or National Diabetes Data Group criteria state a prevalence from 2.2% to 37.9%, while the IADPSG criteria indicate a range of 3.5% to 38.6% (*Bilous et al., 2021*; *Wang et al., 2024*). Currently, the OGTT is considered the gold standard for screening and diagnosing GDM (*Zeck et al., 2007*).

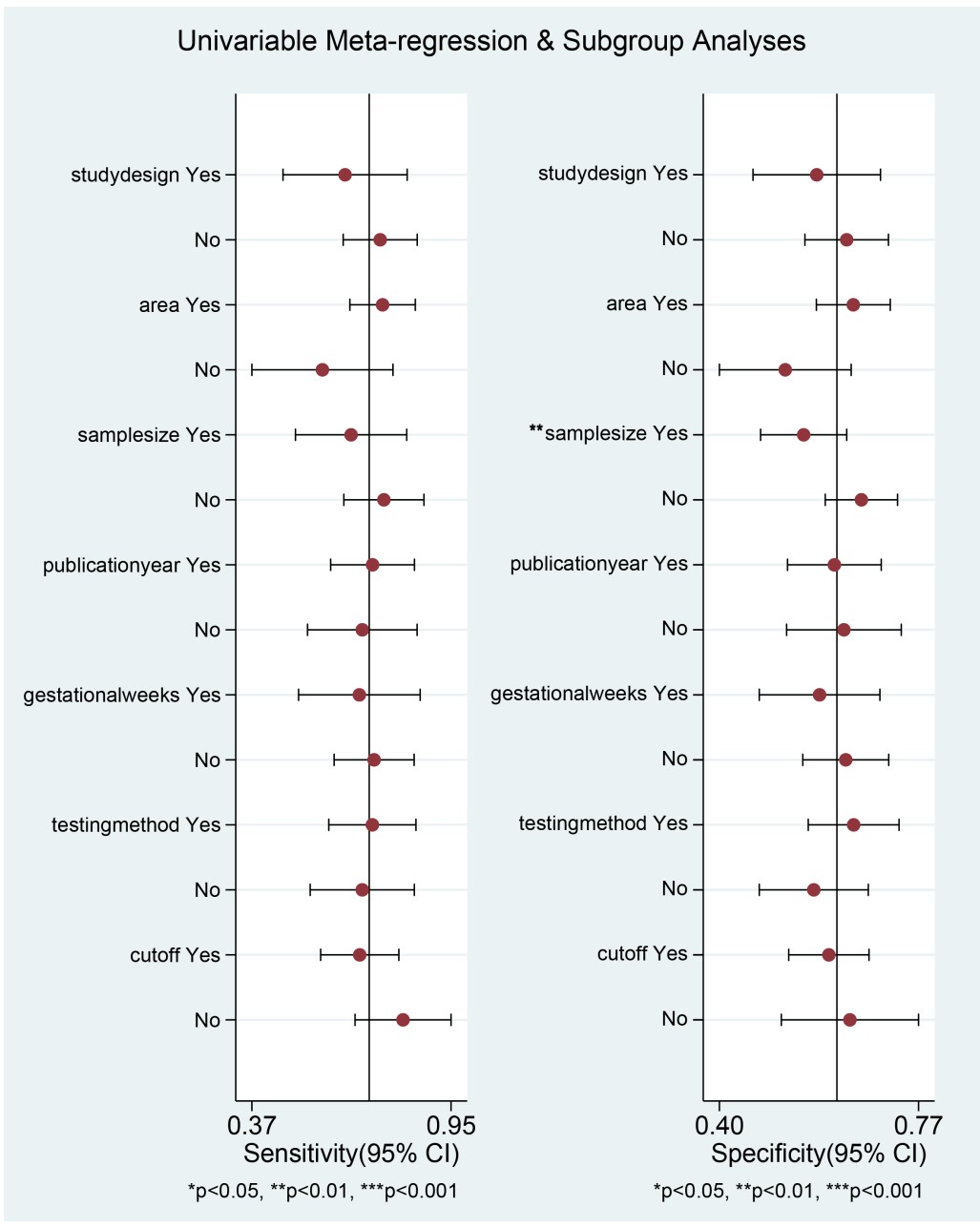

**Figure 5** Univariable meta-regression and subgroup analysis of study design, area, sample size, publication Year, gestational weeks, testing method, cut-off. *$p < 0.05$; **$p < 0.01$; ***$p < 0.001$.

However, this method has limitations, including complex procedures, time consumption, and potential burdens on pregnant women. Consequently, identifying more convenient and accurate biomarkers is significant for early detection and intervention of GDM. Recent research has explored various potential biomarkers, such as PAPP-A and placental IGF (*Leeson, Odendaal & Quenby, 2023*; *Nurdiati et al., 2024*).

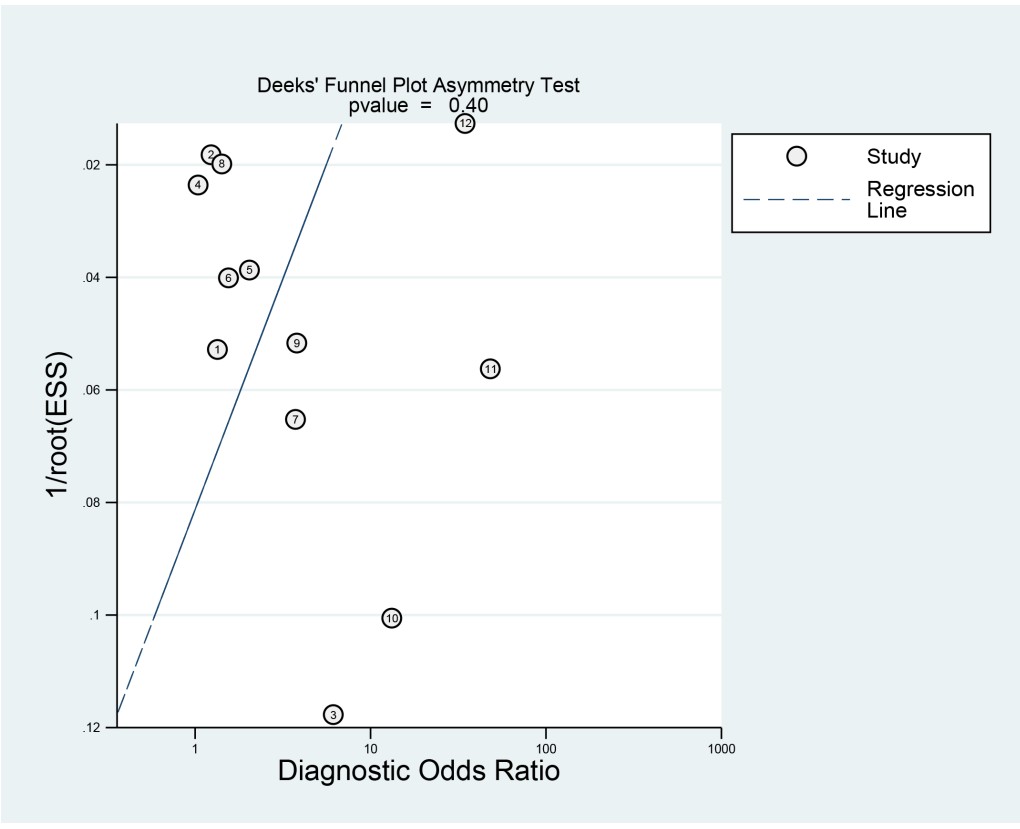

**Figure 6** Publication bias of Deeks' funnel plot asymmetry test.

Since PAPP-A was initially detected in maternal serum, the research discovered that this enzyme (*Conover & Oxvig, 2023*), which required zinc for its activity, was crucial in cleaving specific IGFBPs (*Conover et al., 2004*). It primarily targets IGFBP-4, leading to enhanced levels of free IGF-I available for biological behaviors (*Giudice et al., 2002*; *Yu et al., 2025*). PAPP-A regulates IGF bioavailability by proteolytically hydrolyzing IGFBP-2, -4, and -5 (*Fagali Franchi et al., 2024*). The physiological state of maternal adipose tissue and systemic glucose regulation influence IGF availability (*Sessions-Bresnahan, Heuberger & Carnevale, 2018*). Reduced expression of PAPP-A results in diminished hydrolysis of IGFBPs, consequently decreasing IGF bioavailability. This mechanism is more pronounced during pregnancy to satisfy the increased requirement for bioactive IGF (*Gude et al., 2024*; *Harboe et al., 2024*). Research revealed that serum PAPP-A levels were positively correlated with glycated hemoglobin concentrations and homeostatic model assessment for insulin resistance (HOMA-IR) in GDM patients (*Ku et al., 2024*; *Choi, Duan & Bai, 2025*). This suggests that PAPP-A could elevate blood glucose levels by influencing the insulin signaling pathway. Although PAPP-A shows promise for aiding in GDM diagnosis, it is not yet sufficiently effective to replace the current standard method, the OGTT. In this study, the AUC value of 0.70 indicated that PAPP-A had a relatively modest diagnostic accuracy, which may compromise its reliability as a sole clinical diagnostic marker. Additionally,

specialized equipment and technical expertise are required for PAPP-A measurement, which limits its broad application in primary care settings. Therefore, PAPP-A might be more suited for preliminary screening in high-risk populations rather than serving as a routine diagnostic tool. Subgroup analysis indicated that sample size was likely the main source of heterogeneity ($p < 0.05$). Hence, future studies with larger sample sizes are required to improve the stability and reliability of our results. The QUADAS-2 tool demonstrated an overall high quality. Under the PRISMA guidelines, this meta-analysis significantly enhanced the reliability of the pooled results, surpassing that of individual studies. However, several limitations were noted. First, the search was confined to English-language databases. Second, the limited number of included studies ($n = 12$) could lead to an insufficient sample size and potential bias. Third, variations in sample sources and population characteristics across studies may introduce bias. Finally, the cut-off value used for identifying GDM was inconsistent.

These limitations should be tackled in future research to substantiate the utility of PAPP-A in GDM diagnosis.

## CONCLUSION

This study evaluates the diagnostic accuracy of PAPP-A in GDM at an early stage. The results indicate that PAPP-A has limited capacity to predict individuals at risk for GDM. The current diagnostic effectiveness of PAPP-A alone is insufficient for clinical application. Future research should improve detection methods and establish standardized critical thresholds for PAPP-A. Additionally, the potential value of combining PAPP-A with other biomarkers to enhance GDM diagnostic performance warrants further exploration. Based on our findings, PAPP-A should not be used as a standalone screening or diagnostic test for GDM at present.

**List of Abbreviations**

| | |
|---|---|
| **PAPP-A** | pregnancyassociated plasma protein A |
| **GDM** | gestational diabetes mellitus |
| **AUC** | area under the curve |
| **TP** | true positive |
| **FP** | false positive |
| **FN** | false negative |
| **TN** | true negative |
| **OGTT** | oral glucose tolerance test |
| **IADPSG** | International 57 Association of Diabetes and Pregnancy Study Groups |
| **IGF** | insulin-like growth factor |
| **PRISMA-DTA** | Preferred Reporting Items for Systematic Reviews and Meta-Analyses of Diagnostic Test Accuracy |
| **CI** | confidence interval |
| **PLR** | positive likelihood ratio |
| **NLR** | negative likelihood ratio |

| DOR | diagnostic odds ratio |
| MoM | median multiples of the median |
| PlGF | placental growth factor |
| GCT | glucose challenge test |

### Funding

The Jilin Scientific and Technological Development Program (YDZJ202501ZYTS182); Jilin Province Health Science and Technology Capacity Improvement Project (2024A080). The funders had no role in study design, data collection and analysis, decision to publish, or preparation of the manuscript.

### Grant Disclosures

The following grant information was disclosed by the authors:
The Jilin Scientific and Technological Development Program: YDZJ202501ZYTS182.
Jilin Province Health Science and Technology Capacity Improvement Project: 2024A080.

### Competing Interests

The authors declare there are no competing interests.

### Author Contributions

- Yuge Gao conceived and designed the experiments, performed the experiments, analyzed the data, prepared figures and/or tables, authored or reviewed drafts of the article, and approved the final draft.
- Jihong Chen performed the experiments, prepared figures and/or tables, and approved the final draft.
- Jia Mi conceived and designed the experiments, authored or reviewed drafts of the article, and approved the final draft.

### Data Availability

This is a systematic review/meta-analysis.

### Supplemental Information

Supplemental information for this article can be found online at http://dx.doi.org/10.7717/peerj.19825#supplemental-information.

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
