# Peer review of "Pregnancy-associated plasma protein A in maternal serum for predicting early gestational diabetes mellitus: a systematic review and meta-analysis"

_PeerJ, doi:10.7717/peerj.19825_

## Round 0.1 · original submission · Minor Revisions

- The manuscript should clarify if data and analysis scripts are available for replication.
- While subgroup analysis was performed, further meta-regression could strengthen the assessment of heterogeneity sources.
- The study notes variability in cut-off values; explicit discuss measurement techniques and standardization approaches, if possible
- Revise figure legends for completeness (incl. guiding the readers towards key observations)
- check for typos, including in figures
- define acronyms where appropriate (incl. tables)

**Language Note:** The review process has identified that the English language must be improved. PeerJ can provide language editing services - please contact us at [email protected] for pricing (be sure to provide your manuscript number and title). Alternatively, you should make your own arrangements to improve the language quality and provide details in your response letter. – PeerJ Staff

·

Basic reporting

1. Summary Assessment
Strengths:
- The article provides a comprehensive meta-analysis evaluating the role of PAPP-A in GDM diagnosis.
- Methodologically robust, adhering to PRISMA-DTA guidelines and registered in PROSPERO.
- Subgroup analyses were conducted to identify sources of heterogeneity, and publication bias was assessed
- Results are balanced, with a discussion supporting the clinical utility of PAPP-A in combination with other biomarkers.

Limitations
- High heterogeneity among included studies, which may limit generalizability.
- Insufficient exploration of other heterogeneity sources (e.g., geographic diversity, cutoff thresholds).
- Minor language errors and redundancies (e.g., "GDM diagnosis has become complex and costly" is repeated).

Experimental design

no comment

Validity of the findings

1. Heterogeneity Analysis:
- While the random-effects model is appropriate for high heterogeneity, additional subgroup analyses are needed (e.g., geographic region, GDM diagnostic criteria, cutoff standardization).
- Suggestion: Add subgroups for geographic regions (Asia vs. Europe) and GDM diagnostic protocols.

2. Clinical Relevance Clarification:
- The AUC of 0.67 suggests limited diagnostic accuracy for PAPP-A alone, but the clinical implications of PLR (1.7) and NLR (0.57) are not fully discussed

Additional comments

- Key Contribution: Provides valuable insights into non-invasive biomarkers for GDM diagnosis.
- Future Research: Suggest exploring PAPP-A in combination with HbA1c or insulin resistance markers.

This review highlights the manuscript’s strong foundation while emphasizing the need for revisions to improve clarity and robustness. The authors are strongly encouraged to address these critiques to enhance the impact of their work.

Reviewer 2 ·

Basic reporting

The manuscript addresses an important clinical and diagnostic issue. While it is concise, it requires language editing to improve clarity and correctness. There are missing spaces within the citations, which should be corrected to ensure proper formatting (lines 50, 53, 54, 60, 64, etc.) In the 3.2 "Pooled effect size" section, all abbreviations should be expanded to their full forms for clarity and consistency.
The manuscript adheres to guidelines for conducting a meta-analysis; however, the figures need more attention and improvement in terms of clarity and presentation.
Table 1 should include explanations for all abbreviations used (TP, FP, FN, TN). Additionally, Table 2 is not titled and requires proper labeling.

Experimental design

no comment

Validity of the findings

In the case of conclusion, AUC = 0.67 is a rather low value, suggesting limited diagnostic utility. It is worth naming it explicitly (e.g. "limited" instead of "moderate"). The manuscript does not sufficiently address the clinical implications or potential utility of the results, which limits its translational relevance. In my opinion, based on the results, it is worth adding that standalone PAPP-A testing should not be used as a screening or diagnostic test for GDM at this time. Further studies should aim to establish uniform cutoff values and investigate the potential of combining PAPP-A with additional early-pregnancy biomarkers to improve diagnostic performance.

---

## Round 0.2 · accepted · Accept

Dear authors, I am now accepting your manuscript for publication in PeerJ. Congratulations! Thank you for your submission and work throughout this process.

Reviewer 2 ·

Basic reporting

no comment

Experimental design

no comment

Validity of the findings

no comment

Additional comments

Thank you for the opportunity to review the revised version of the manuscript. I would like to confirm that all of my previous comments and suggestions have been thoroughly addressed by the authors. The revised manuscript shows significant improvement and meets the standards of the journal. Therefore, I recommend the manuscript for publication in its current form.